# fNIRS Complexity Analysis for the Assessment of Motor Imagery and Mental Arithmetic Tasks

**DOI:** 10.3390/e22070761

**Published:** 2020-07-11

**Authors:** Ameer Ghouse, Mimma Nardelli, Gaetano Valenza

**Affiliations:** 1Bioengineering and Robotics Research Center E Piaggio, Università di Pisa, 56123 Pisa, Italy; m.nardelli@ing.unipi.it (M.N.); g.valenza@ing.unipi.it (G.V.); 2Department of Information Engineering, Università di Pisa, 56123 Pisa, Italy

**Keywords:** fNIRS, entropy, complexity analysis, nonlinear analysis, brain dynamics, mental arithmetics, motor imagery

## Abstract

Conventional methods for analyzing functional near-infrared spectroscopy (fNIRS) signals primarily focus on characterizing linear dynamics of the underlying metabolic processes. Nevertheless, linear analysis may underrepresent the true physiological processes that fully characterizes the complex and nonlinear metabolic activity sustaining brain function. Although there have been recent attempts to characterize nonlinearities in fNIRS signals in various experimental protocols, to our knowledge there has yet to be a study that evaluates the utility of complex characterizations of fNIRS in comparison to standard methods, such as the mean value of hemoglobin. Thus, the aim of this study was to investigate the entropy of hemoglobin concentration time series obtained from fNIRS signals and perform a comparitive analysis with standard mean hemoglobin analysis of functional activation. Publicly available data from 29 subjects performing motor imagery and mental arithmetics tasks were exploited for the purpose of this study. The experimental results show that entropy analysis on fNIRS signals may potentially uncover meaningful activation areas that enrich and complement the set identified through a traditional linear analysis.

## 1. Introduction

Functional near-infrared spectroscopy (fNIRS) is a noninvasive technique that has found success in analyzing brain function through the lens of metabolic processes and neurovascular coupling [1,2]. Common methods found in the literature analyze fNIRS signals with the assumption that an underlying linear system generated their time series [3]. Though these approaches may find success in some domains, linearity is an ideal assumption when investigating brain physiology. In fact, many physiological systems exhibit nonlinear behavior, meaning there can be further interaction between variables in a system beyond a superposition effect while also having dynamics that the system sub-components may not show. Beyond nonlinearity, physiological systems may exhibit complex dynamics as a result of feedback loops that arise from homeostasis regulation with consequent extreme sensitivity to the system state condition [4,5,6].

Prior literature has shown that nonlinearities are particularly present in the brain and its related metabolic processes. Functional magnetic resonance imaging (fMRI) and fNIRS data were demonstrated to follow a nonlinear saturating impulse response model [7], and physiological models of cerebral blood flow dynamics include complex feedback loops between ion channels, metabolism, energy demand, and oxygenation [8]. Furthermore, dynamics of the intrinsic parameters, such as the electrophysiological process that drives neurovascular coupling, also exhibit nonlinear and complex behavior [9,10].

Such nonlinearities found in metabolic processes imply that standard linear models and metrics quantifying linear dynamics defined in the time and frequency domains may potentially underrepresent the physiological processes sustaining functional activity. To this end, entropy can be a powerful tool to characterize a system’s regularity or complexity [11]. When applied to the topology of attractors describing a dynamical system in phase space, entropy leads to a robust estimation of regularity of state space evolution, also known as the Kolmogorov–Sinai metric [12]. By exploiting Takens’ theorem and the concept of characterizing an attractor through its topological entropy, several algorithms have been developed to find a value that converges to the Kolmogorov–Sinai entropy metric for regularity. Such algorithms include sample entropy (SampEn) [13] and fuzzy entropy (FuzzyEn) [14], which are able to characterize a system’s regularity at a single time scale level [15]. On the other hand, metrics, such as distribution entropy (DistEn) [16], have been shown to provide complexity estimates of the system under study.

While entropy analysis has been a widely investigated tool for studying electrophysiological signals, there is a dearth of studies regarding entropy applied to metabolic processes, as observed in fNIRS signals. Permutation entropy, i.e., entropy of a time series from an ordinal transform on the continuous data [17,18], has been exploited by Gu et al. to investigate the complexity of fNIRS signals in children affected by attention deficit disorder during working memory tasks [19]. Furthermore, Jin et al. investigated permutation entropy to analyze differences in experts and novices solving science problems [20]. Studying frontal cortex fNIRS signals, SampEn was suggested as a biomarker for Alzheimer’s disease diagnosis [21,22,23], and Angsuwatanakul et al. [24] investigated the effects of working memory experiments on SampEn estimated from fNIRS series. Also, though applied as an information theoretic approach to investigate linear effects in fNIRS rather than analyze topological entropy in phase space, differential entropy has been investigated in Keshmeri et al. as a biomarker that preserves variational information in the assessment of working memory [25,26].

Although there is literature for entropy applied to fNIRS signals, there has yet to be an analysis of its regularity and complexity during standard cognitive load tests, such as motor imagery and mental arithmetics. Besides, previous studies using entropy were not performed using a time stamped controlled block design protocol. Thus, it is not yet clear how well entropy as an estimate works when activity is controlled in time. Furthermore, a comparison with standard methods deserves scrutiny. To overcome these oversights, this study aims to uncover SampEn, FuzzyEn, and DistEn estimates of hemoglobin, deoxyhemoglobin, and total hemoglobin in mental arithmetics and motor imagery experiments in order to perform a comparison with traditional methods in fNIRS signals analysis. Concretely, we hypothesize that by considering nonlinear and complex characterizations of metabolic processed observed in fNIRS signals, more information, as expressed by cortical activity correlates, can be gleaned regarding physiological and psychophysiological phenomena than what can be considered using only linear analyses. For the purpose of this study, we used an open access dataset provided by Shin et al. [27], whose details on methodology and results follow below.

## 2. Materials and Methods

### 2.1. Block Design

The dataset used in this study is openly available and fully described in [27]. Briefly, twenty-nine subjects (aged 28.5 ± 3.7, 15 females) were involved in the experiment. Left and right hand motor imagery constituted one set of trials performed, and the other set of trials were baseline and mental arithmetics. There were three trials of each of the aforementioned experiments per subject. fNIRS and electroencephalography (EEG) series were acquired simultaneously during the whole duration of the experiment using 30 EEG channels and 36 fNIRS channels, and the sampling rate for fNIRS signals was 10 Hz. The 36 fNIRS channels were resolved from a set of 14 sources to 16 detectors matching as illustrated in Figure 1.

The experimental protocol began with a 60 s rest, after which subjects were presented an instruction (either a “←” or “→” for motor imagery experiments, and either a “-” or an arithmetic task in comparing baseline vs mental arithmetic) on the screen telling them which task were to be performed. Afterwards, the individual performed the task for 10 s, with a subsequent 15 s rest before the next task. After 20 repetitions of these instructions and tasks, a 60 s ending rest was performed. Mental arithmetic/baseline trials were performed independently from motor imagery trials.

### 2.2. Hemoglobin Extraction from fNIRS Signals

In continuous wave fNIRS acquisitions, light radiations from two different wavelengths are used to create a system of equations that can resolve hemoglobin content. These wavelengths are generally chosen to be in the range of the physiological window where water and hemoglobin absorption is particularly low (650 nm to 1350 nm). To this extent, the “modified Beer–Lambert law” provides a mathematical expression relating absorption measured with a detector and the concentration of a chromophore as seen in Equation (Equation 1) [28]:(1)$$\mu_{a}\left(\lambda ,t\right)=log\left(\frac{I_{o}\left(\lambda \right)}{I(\lambda ,t)}\right)=\sum_{i=0}^{n}c_{i}\left(t\right)\epsilon_{i}\left(\lambda \right)\rho DPF+G$$
where $\mu_{a}$ is the absorption coefficient at a given wavelength λ and time *t*, $I_{o}$ is incident light intensity, *I* is the detected intensity that changes with time, *c* are the chromophore concentrations of interest, ρ is the separation between a light source and detector, DPF is the correction factor for a best estimate of a light path through a tissue, and G is the loss of light due to scattering. In a continuous wave setting, differential concentrations Δc, related by differential absorption $\Delta \mu_{a}$, are the parameters that are analyzed in the fNIRS signals. This allows for a significant simplification of the expression above when assuming that scattering loss is a constant in time, yielding differential concentrations that can be resolved through a simple linear system of equations given multiple wavelengths, as seen in Equation (Equation 2):(2)$$\Delta \mu_{a}\left(\lambda ,t\right)=log\left(\frac{I_{b}\left(\lambda \right)}{I(\lambda ,t)}\right)=\sum_{i=0}^{n}\Delta c_{i}\left(t\right)\epsilon_{i}\left(\lambda \right)\rho DPF$$
where $I_{b}$ is the intensity detected at a baseline of interest.

From the modified Beer–Lambert law, the differential concentration of deoxyhemoglobin can be retrieved by choosing two wavelengths on opposing sides of the isobestic point of the absorption spectra of oxyhemoglobin and deoxyhemoglobin and solving a linear system of equations. In the methods of Shin et al., 760 nm and 850 nm were used as wavelengths. In this study, an evaluation on such differential concentrations as well as total Hb during activity will be performed.

### 2.3. fNIRS Data Preprocessing

Figure 2 shows an overview of the preprocessing pipeline. The signal was transformed from optical densities into Hb and HbO using the modified Beer–Lambert law. For the modified Beer–Lambert law, the first 60 s were considered as a baseline, corresponding to the resting state. A first Butterworth lowpass filter with a cutoff frequency at 0.6 Hz and filter order 6 was applied to fNIRS data to highlight the hemodynamic response. This was considered as 0.6 Hz is the upper end of cut-off frequencies used in literature [29]. This is significant for preserving the full dynamics of hemoglobin, including the high frequency components, which can uniquely affect the topology of the attractor in phase space and render different estimates of entropy. In hand, we must accept the risk of physiological phenomena, such as Mayer waves, contaminating the entropy estimates. A second band-pass filter with cutoffs 0.8 Hz and 2 Hz and filter order 6 was used to capture pulsatile dynamics of hemoglobin [30]. Afterwards, a wavelet filtering approach was used to further reduce noise, particularly related to motion artifacts, in the oxy- and deoxyhemoglobin signals [31]. This wavelet filtering approach works by decomposing the time series into nine levels using a daubechies five mother wavelet, subsequently thresholding detail coefficients that have low probability (*p* < 0.1) given the detail coefficients are sampled from a normal distribution. After the wavelet filter, the time series were separated into epochs representing blocks of activity. Each channel at each activity block was differentially referenced to the mean of the previous 5 s of said channel. The data was then further processed to extract features such as entropy and mean values of hemoglobin.

### 2.4. Entropy Analysis

The entropy metrics SampEn, FuzzyEn, and DistEn were extracted as regularity and complexity characterizations of fNIRS data. For each fNIRS signal (Hb, HbO, and total hemoglobin) and for the multivariate embedding derived from a concatenation of the Hb and HbO embedding, the optimal time delay was chosen as the first zero of the autocorrelation while the optimal embedding dimension was found using the false nearest neighbours algorithm [32].

To create the embeddings, we started from a time series x(t) of *N* samples. Having determined the time lag τ and embedding dimension *m*, the states $X_{t}^{m}$ of the reconstructed attractor can be represented in vector form as follows:(3)$$X_{t}^{m}=\left\{x\left(t\right),x\left(t+\tau \right),\ldots x\left(t+\left(m-1\right)\tau \right)\right\}$$

When reconstructing an attractor using several variables, i.e., the concatenated attractor (Concat), the above expression is modified in the following way:(4)$$X_{t}^{m}=\left\{x\left(t\right),x\left(t+\tau \right),\ldots x\left(t+\left(m-1\right)\tau \right),y\left(t\right),y\left(t+\tau \right),\ldots y\left(t+\left(m-1\right)\tau \right)\right\}$$

From the reconstructed attractor, entropy estimates may be computed. For SampEn, the estimate can be obtained, as follows:(5)$$SampEn=-log(\frac{\sum_{i=1,i\neq j}^{N-m}\frac{1}{N-m-1}\mathrm{number}\;\mathrm{of}\left|X_{i}^{m+1}-X_{j}^{m+1}\right|<R}{\sum_{i=1,i\neq j}^{N-m}\frac{1}{N-m-1}\mathrm{number}\;\mathrm{of}\left|X_{i}^{m}-X_{j}^{m}\right|<R})$$
where *R* refers to a user set deviance of states to binarize the distance metric. In this study, it is set to $0.2*\sigma_{x}$, a setting widely used in previous studies with theoretical justifications, where $\sigma_{x}$ is the standard deviation of the considered time series [33,34].

FuzzyEn uses a fuzzy membership function instead of the heaviside function to calculate the correlation integral. In this study, we employed an exponential function, as follows:(6)$$\phi^{m}=\frac{1}{N-m}\sum_{i=1}^{N-m}\frac{1}{N-m-1}\sum_{j=1,j\neq i}^{N-m}e^{\frac{\left(-\right|X_{i}^{m}-X_{j}^{m}{\left|\right)}^{K}}{R}}$$

The value of *K* was set to 2. Afterwards, FuzzyEn can be derived as the ratio between the above fuzzy function with the result of a fuzzy function of an order greater [35].
(7)$$FuzzyEn=-log(\frac{\phi^{m+1}}{\phi^{m}})$$

DistEn is less dependent on parameter selection in comparison to FuzzyEn and SampEn, given that the parameter *R* is no longer required. A histogram is constructed from the distance matrix, and the Shannon entropy of the empirical probability density function is computed. To make the algorithm faster, we extracted the upper triangle of the distance matrix, as it should be symmetrical, meaning that lower triangle contains redundant information. Additionally, the diagonal is removed from the entropy estimate as it should be a zero vector when considering that the self-similar distance is zero. Bin size was estimated by using Scott’s method [36].

### 2.5. Statistical Analysis

A bootstrapped third moment test was performed with linear time series surrogate samples generated by an amplitude adjusted Fourier transform of the original time series and phase scrambling in order to test the null hypothesis that the original time series was generated from a linear system [37]. Two-hundred surrogate series were generated in order to determine a p-value, with the third moment calculated for τ=1 lag as illustrated in the following equation [38].
(8)$$t^{c3}\left(\tau \right)=<x_{k}\cdot x_{k+\tau }\cdot x_{k+2\tau }>$$

The percentage of significant time series for each channel and each parameter, either Hb, HbO, or total Hb, are shown in the results. Significance is determined using an α=0.05.

After ascertaining nonlinearity, further non-parametric tests were performed on entropy and mean hemoglobin results when considering the non-Gaussian distribution of the metrics. Friedman non-parametric statistical tests for paired data were performed in order to determine whether repetitions of activities in each trial were significantly different. Afterwards, a Friedman test was applied using a median summary statistic over trials to compare significant cortical areas of activation between the four tasks (i.e., baseline, mental arithmetic, left hand, and right hand motor imagery). Multiple comparison tests were then performed between pairs of tasks using Wilcoxon signed rank tests for paired data, and the statistical significance was set to 0.05 when considering a Bonferroni correction rule over the four different activity comparisons.

Group-wise and channel-wise multiple comparison results for each metric are displayed using both p-value topographic maps and topographic maps displaying Δ value differences between tasks for a given metric. Cortical regions in the topographic maps that are not covered by the optodes, as seen in Figure 1, are inferred using a bilinear interpolation.

## 3. Results

### 3.1. Nonlinearity Test

As illustrated in Figure 3, an analysis of the third moment for each time series in Hb, HbO, and total hemoglobin demonstrates that the majority of time series exhibits nonlinear behavior, rejecting the null hypothesis that a linear system generated the time series.

### 3.2. Analysis of Repetitions within Tasks

Through the Friedman statistical test on repetitions, it can be seen by Table 1 that we were able to accept the null hypothesis that there were no significant differences between the repetitions for either mental arithmetic, left hand imagery, right hand imagery, or baseline when using any of the statistics of mean, SampEn, or DistEn over any set of hemoglobin time series representation. On the other hand, we could reject the null hypothesis for the FuzzyEn comparisons in the case of using total hemoglobin time series and the multivariate topology reconstructed from both oxyhemoglobin and deoxyhemoglobin.

Given this result, subsequent post-hoc analyses focused on mean estimates, SampEn, and DistEn for each time series. Furthermore, when considering that the repetitions of these metrics did not show significant differences, the median value of each estimate over repetitions was used as a summary statistics for further inter-subject analyses.

### 3.3. Between-Task Statistical Analysis

Cortical areas with significant statistical differences between baseline, mental arithmetic, right hand, and left hand motor imagery tasks according to a Friedman test analysis on mean, SampEn and DistEn analyses can be seen in Figure 4. Estimates on the oxyhemoglobin signal showed overlapping areas of significance between mean estimate and both entropy estimates in the occipital regions. On the other hand, DistEn estimates on deoxyhemoglobin signal had significant changes between tasks over the somatosensory cortex that were not exhibited in the mean estimate. For the total hemoglobin signal, both SampEn and DistEn unraveled further information that mean estimates could not, where DistEn exhibited significant changes in the occipital area, and SampEn exhibited changes in the parietal area. In the concatenated topology, DistEn and SampEn exhibited different subsets of cortical activations.

### 3.4. Multiple Comparison Analysis

Figure 5 shows cortical areas that were associated with significant statistical differences between mental arithmetic activity and baseline activity for a given estimate according to Wilcoxon non-parametric tests. When analyzing oxyhemoglobin, SampEn displayed regions in the occipital cortex that were not highlighted by the mean estimates; DistEn did not seem to add further information. From deoxyhemoglobin signal analysis, DistEn uncovered significant changes over the left occipital region that were unobserved in the mean estimate analysis; SampEn did not seem to add new information. On total hemoglobin, significant changes between tasks were found in the right occipital cortex from DistEn, which were unobserved in mean estimates. From the concatenated signal, DistEn displayed information in the parietal cortex that was unobserved by previous analysis. From visual inspection on Figure 5, it seemed that mental arithmetic activity was generally associated with higher mean and a lower irregularity and complexity levels than baseline.

Further Wilcoxon tests were performed to show cortical areas that were associated with a significant statistical difference between left hand motor imagery and baseline for a given estimate, as seen in Figure 6. When analyzing oxyhemoglobin, both DistEn and SampEn showed significant changes between tasks over a larger region than the mean estimate, especially in the right occipital cortex. Furthermore, deoxyhemoglobin activity in the left temporal and sensorimotor cortices was highlighted by both entropies. A total hemoglobin analysis confirmed that DistEn and SampEn highlight further changes that were not seen in a mean estimate analysis. With visual inspection on Figure 6, it appeared that SampEn inversely mapped mean estimate changes over the the frontal, motor, and parietal regions for the oxyhemoglobin signals. For deoxyhemoglobin, higher mean estimates over the right hemisphere were associated with left hand motor imagery activity. SampEn increased over the frontal areas during left hand motor imagery tasks with respect to baseline with no changes over the posterior areas. Changes in total hemoglobin signal seemed similar to oxyhemoglobin.

From Figure 7, another set of Wilcoxon non-parameteric test results can be seen, showing cortical areas that were associated with a significant statistical difference between right hand motor imagery and baseline for a given estimate. While mean estimates were associated with few significant changes between tasks, SampEn and DistEn showed significant differences over several areas, especially in a oxyhemoglobin and total hemoglobin analysis. Particularly, in a oxyhemoglobin analysis, DistEn showed significant changes over the frontal, right, and left occipital areas, which were complemented by further changes over parietal cortices by SampEn. For deoxyhemoglobin signal, complementary parietal activity appeared in DistEn while SampEn changes were a subset of the mean estimates. In the case of total hemoglobin, changes over the sensorimotor and parietal cortices were found using SampEn, while DistEn and mean estimates did not show significant changes between tasks.

Using further visual inspection analysis on Figure 7, the trend appears to be that higher mean, SampEn, and DistEn values over the frontal areas were more associated with right hand motor imagery activity, whereas higher estimate values over the posterior areas were associated with baseline activity. In the case of deoxyhemoglobin, higher mean estimates over the right hemisphere were associated with right hand motor imagery activity. SampEn increased over the frontal areas during right hand motor imagery tasks with respect to baseline with no changes over the central posterior areas. Changes in total hemoglobin signal seemed similar to oxyhemoglobin ones.

Figure 8 shows cortical areas that were associated with a significant statistical difference between right hand and left hand motor imagery for a given estimate according to Wilcoxon non-parametric tests. Complementary left occipital activity was uncovered by DistEn for oxyhemoglobin, while a deoxyhemoglobin analysis using SampEn uncovered unique parietal activity changes between tasks. In the case of total hemoglobin, larger parietal changes were found in DistEn than the mean estimate, while SampEn exhibited changes in the right temporal regions. Visual inspection analysis on Figure 8 shows a trend of left hand motor imagery activity being associated with higher mean, irregularity and complexity levels than right hand motor imagery activity over the frontal areas, while an opposite trend seemed to be observed over the posterior regions. Particularly, changes over the frontal cortex in mean estimates seemed similar to SampEn differences in oxyhemoglobin, while they appeared to be inversely distributed in DistEn. In deoxyhemoglobin, no differences between left and right hand motor images seemed to occur over the posterior regions in SampEn, whereas DistEn appeared to show similar differences as mean estimates.

## 4. Discussion

We investigated changes in fNIRS entropy during mental arithmetics and motor imagery tasks and compared the results with fNIRS standard analysis metrics. Our aim was to test whether entropy analysis could unravel changes in cortical areas that may not be highlighted while using traditional methods that analyze the signal in the time domain. Particularly, we assessed statistical differences in fNIRS signal entropy in four different tasks (baseline, mental arithmetic, right hand, and left hand motor imagery), and compared different entropy metrics—specifically SampEn, FuzzyEn, and DistEn—together with mean value estimates of hemoglobin, deoxyhemoglobin, and total hemoglobin.

Previous studies used entropy estimates in protocols of long time windows with unspecified timing of events in the signal, as in the case of cognitive capacity analysis in Alzheimer’s [21,22,23]. Nevertheless, regularity and complexity analyses of fNIRS signals during standard cognitive load tests, such as motor imagery and mental arithmetics, were not investigated to the best of our knowledge.

Through a test of nonlinearity, we were able to ascertain that the majority of the considered time series demonstrated nonlinear behavior. Nonlinearity testing was necessary for validating whether quantifying the extent of nonlinear behavior could be a value of interest in the analysis of functional activity. To that extent, our analysis corroborated studies performed in the past, such as the evidence demonstrated in Khoa et al., where they performed similar nonlinearity tests [38].

Our study showed that FuzzyEn applied to total hemoglobin and the concatenated attractor from the open dataset demonstrated significant differences between task repetitions (see Table 1); therefore, only SampEn and DistEn were retained for further analyses on fNIRS regularity and complexity at a task level. In fact, this result allowed for subsequent comparison analyses between the four tasks to be performed using a median summary statistic for entropy and the mean estimates over the repetitions rather than considering each repetition independently.

Over the general set of results, complementary areas of functional activity were found in both SampEn and DistEn when compared to mean estimates, as demonstrated in Figure 5, Figure 6, Figure 7 and Figure 8. For example, in the comparison between mental arithmetics and baseline activities in Figure 5, SampEn was able to uncover particular parietal activity in oxyhemoglobin that mean estimates, using any of the three hemoglobin concentrations (Hb, HbO, THb), were unable to resolve. Furthermore, it appears that both entropy estimates are more sensitive to temporal cortex activity, as seen in Figure 6 and Figure 7, when analyzing motor imagery tasks compared to baseline.

Previous studies highlighted hemodynamic changes during mental arithmetic tasks primarily over the bilateral intraparietal, inferior temporal, and dorsal prefontal sites [39,40]. SampEn and DistEn were both successful in recovering those activity areas as demonstrated in Figure 5. Particularly, SampEn applied to oxyhemoglobin showed changes over parietal structures while deoxyhemoglobin revealed changes over frontal cortical sites. With DistEn, the concatenated series displayed changes over both parietal, frontal, and temporal activity. However, the mean estimates were not able to uncover the parietal cortex changes, but instead were only sensitive to frontal cortex and temporal cortex activity. In the case of mental arithmetics, these results suggest that entropy estimates may be more sensitive to cortical hemodynamic changes than mean estimates given the sample size available. This may be due to the additional quantification of nonlinear and complex dynamics provided by entropy analysis. Where linear effects subside or may not be as significant, nonlinear, and complex behavior may still persist. This could be explained by models that demonstrate short term stimuli resulting in nonlinear behavior in the hemodynamic response [7]. Speculatively, stimuli may become less frequent or last for shorter durations when a subject experiences fatigue from a protocol or has become habituated.

In light of motor imagery tasks, Figure 8 demonstrates that we were able to find activity areas in the expected sensorimotor cortex while using either entropy or mean estimate analysis. Explicitly, both DistEn applied to deoxyhemoglobin and SampEn applied to the concatenated attractor unraveled these expected changes. Furthermore, we observed a lateralization effect in DistEn applied to oxyhemoglobin and the concatenated attractor, as well as SampEn applied to oxyhemoglobin. These results are in accordance with previous findings [41]. This suggests the presence of complementary information supplied by regularity and complexity analysis on fNIRS series. In light of these significant results, it is important to mention that a bilinear spatial interpolation was performed on the topographic maps as mentioned in Section 2.5, thus there could be errors in drawing the true cortical location of activity. It would be important to use a higher density fNIRS cap in future experiments in order to better pinpoint the true cortical location of a specific activity.

The success of entropy estimates in unraveling complementary areas was particularly surprising when considering that the experiments studied here were tailored to leverage strong activations that arise from a saturating superposition effect, i.e., linear superposition. As mentioned above, it is possible that short-term stimuli were introduced when either the subject became habituated or fatigued. Furthermore, there may also be significant oscillatory behaviors that contribute to the observed nonlinearities observed in the hemodynamic signal that mean value analysis can not detect. For example, changes in pulsatility in the microvessels that arise from cardiac pulses and physical properties of the microvessels may nonlinearly affect the oxygen extraction from the capillaries to the tissue [10].

As has been mentioned in the introduction, biological systems exhibit a vast array of feedback and compensatory loops in order to regulate homeostatic behavior at a neurolobiological level [4,5,6]. This knowledge brings light to the significance of the study we have presented in leveraging the information in phase space that this complex system projects in the fNIRS time series. However, a clear limitation of the study is that it is purely exploratory, rather than explanatory for the neurobiological activity that underlies the complex system the entropy estimates assess. Nonetheless, this study holds a beacon for future research to investigate the intrinsic complex neurobiological correlates that comprise activity in mental arithmetic and motor imagery tasks.

A natural extension of this study in the future can be to apply fNIRS regularity and complexity analysis to block-free paradigms, such as a clock drawing test [42], or tests that stimulate more complex dynamics related to emotional response [43,44,45]. Because SampEn was not applied using a multiscale algorithm, future studies can also investigate fNIRS dynamic activity while using a multiscale entropy analysis. Such sophisticated methodology may further highlight complex changes that may be induced by activity on different time scales, such as cardiac pulsatility, arterial blood pressure induced mayer waves, or other nonlinearities driving the hemodynamic response [10,46,47]. Furthermore, in future studies, a dataset using an fNIRS system that includes short source-detector separation channels can be analyzed to regress out artifacts due to skin-blood flow induced changes in the fNIRS signals.

## 5. Conclusions

A novel investigation into the analysis of entropy in metabolic processes measured by fNIRS on controlled block design experimental protocols was presented in this study. We conclude that entropy may uncover areas that yield neuronal correlates and that agree with traditional methods of analyzing neuronal correlates while also providing novel complementary areas not seen in mean estimates. Furthermore, entropy estimates seemed to exhibit greater sensitivity with sample size to activity than mean estimates in mental arithmetics. These results shed light on not only the validity, but also the efficacy of using entropy to investigate functional neural activations.

## Figures and Tables

**Figure 1 entropy-22-00761-f001:**
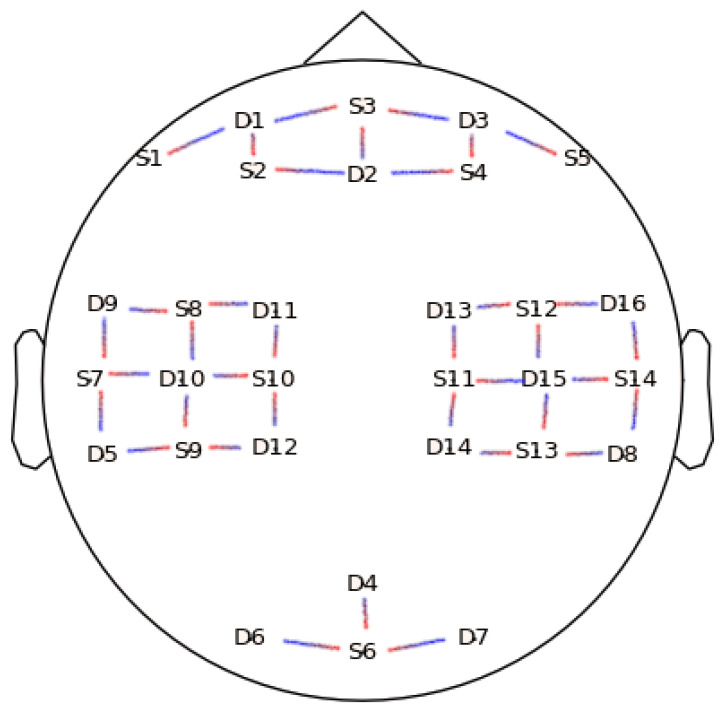
Position of the Optodes. Positions labeled with “D” refer to detectors while positions labeled with “S” are sources. The lines demonstrate coupling between sources and detectors.

**Figure 2 entropy-22-00761-f002:**
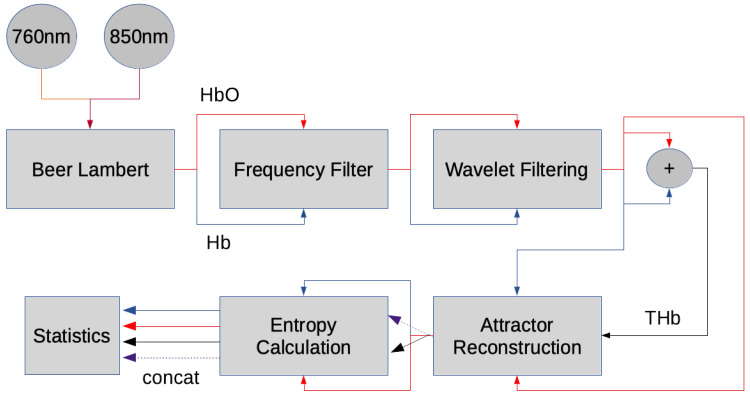
Pipeline for processing functional near-infrared spectroscopy (fNIRS) data.

**Figure 3 entropy-22-00761-f003:**
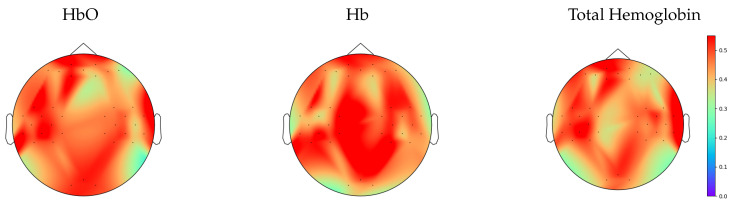
Topographic maps from channel-wise third moment tests displaying the fraction of time series from each channel having statistical significance, where the colorbar indicates the value of the fraction.

**Figure 4 entropy-22-00761-f004:**
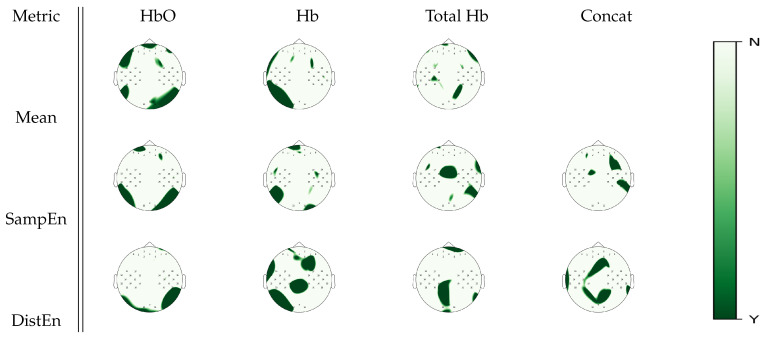
*p*-value topographic maps from channel-wise Friedman tests displaying significant statistical differences between all tasks in the experimental protocol (baseline, mental arithmetic, right hand, and left hand motor imagery). Y (green) areas indicate where we could reject the null hypothesis that activity was the same in all the tasks, whereas N (white) areas indicate where we could not reject the null hypothesis.

**Figure 5 entropy-22-00761-f005:**
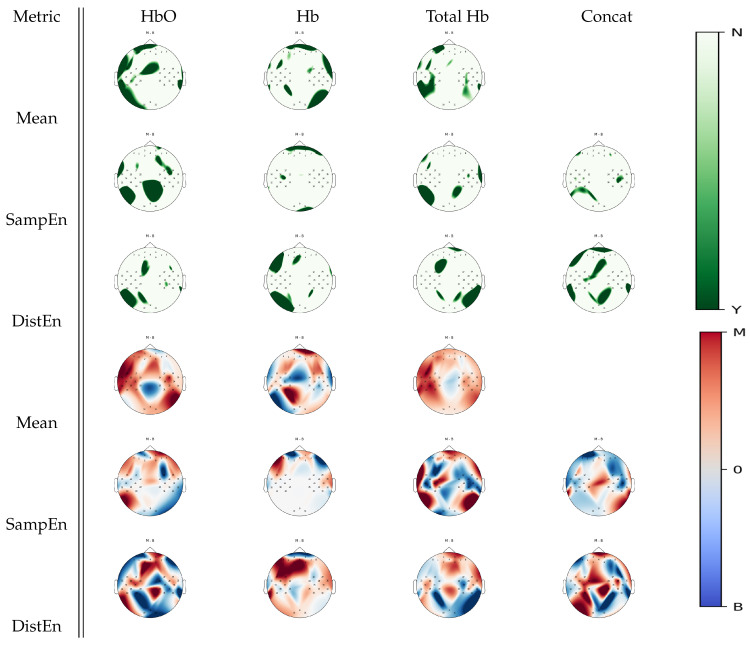
*p*-value topographic maps from channel-wise Wilcoxon non-parametric tests displaying significant statistical differences between mental arithmetic activity and baseline activity. Y (green) areas indicate statistically significant changes between tasks, whereas N indicates non-significant changes. The colormap topoplots display estimate differences between baseline (B) and mental arithmetic (M) tasks, with red indicating higher values for mental arithmetic than baseline and blue indicating lower values for mental arithmetic as compared to baseline.

**Figure 6 entropy-22-00761-f006:**
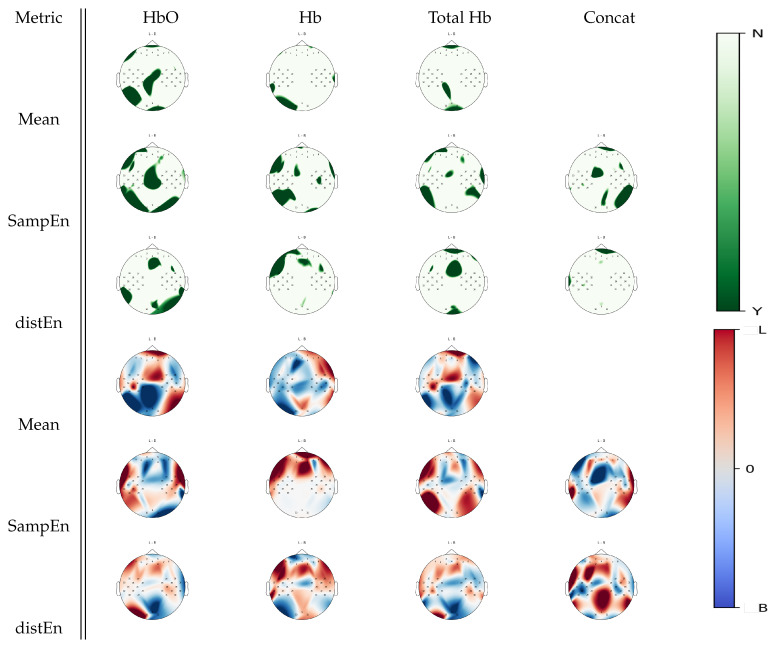
*p*-value topographic maps from channel-wise Wilcoxon non-parametric tests displaying significant statistical differences between left hand imagery activity and baseline activity. Y (green) areas indicate statistically significant changes between tasks, whereas N indicates non-significant changes. The colormap topoplots display estimate differences between baseline (B) and left hand imagery (L) tasks, with red indicating higher values for left hand imagery vs baseline and blue indicating lower values for left hand imagery vs baseline.

**Figure 7 entropy-22-00761-f007:**
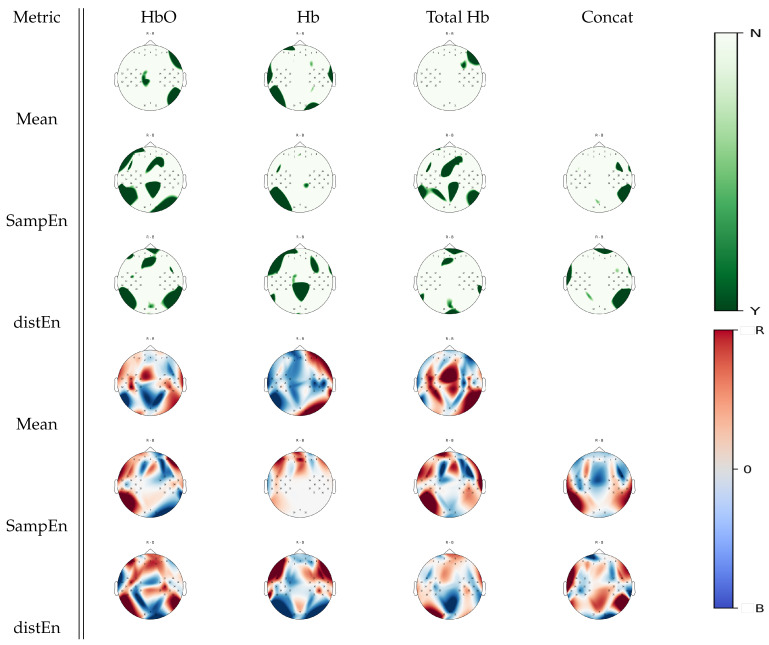
*p*-value topographic maps from channel-wise Wilcoxon non-parametric tests displaying significant statistical differences between right hand imagery activity and baseline activity. Y (green) areas indicate statistically significant changes between tasks, whereas N indicates non-significant changes. The colormap topoplots display estimate differences between baseline (B) and right hand imagery (R) tasks, with red indicating higher values for right hand imagery vs baseline and blue indicating lower values for right hand imagery vs baseline.

**Figure 8 entropy-22-00761-f008:**
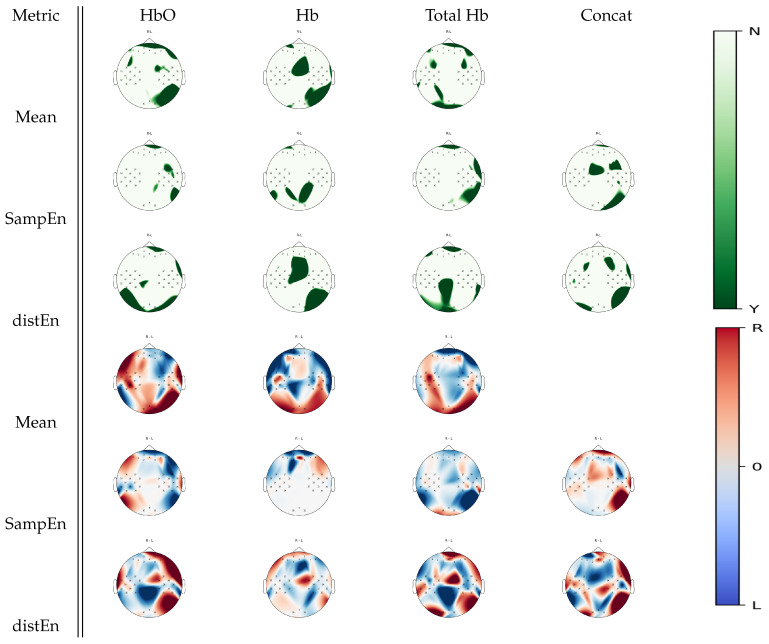
*p*-value topographic maps from channel-wise Wilcoxon non-parametric tests displaying significant statistical differences between left hand imagery and right hand imagery activities. Y (green) areas indicate statistically significant changes between tasks, whereas N indicates non-significant changes. The colormap topoplots display estimate differences between left hand imagery (L) and right hand imagery (R) tasks, with red indicating higher values for right hand imagery than left hand imagery and blule indicating lower values for right hand imagery than left hand imagery.

**Table 1 entropy-22-00761-t001:** Table of statistical power *p*-values from the Friedman analysis. *p*-values are bonferroni corrected. * denotes that using an alpha of 0.01 we must reject the null hypothesis that there were no significant variations between repetitions. This particularly occurs for FuzzyEn in the total and the concatenated case for deoxyhemoglobin.

Metric	Mental Arithmetic	Left Hand Imagery	Right Hand Imagery	Baseline
HbO	0.1735	0.1147	0.0331	0.7383
Hb	0.0870	0.0841	0.1735	0.0039
Total Hb	0.0331	0.2449	0.0965	0.0501
$SampEn_{HbO}$	0.0610	0.1414	0.0976	0.1375
$SampEn_{Hb}$	0.0891	0.2844	0.0101	0.0262
$SampEn_{Total}$	0.2013	0.2528	0.0501	0.0554
$SampEn_{concat}$	0.0408	0.1147	0.0106	0.1735
$FuzzyEn_{HbO}$	0.0934	0.0023	0.0501	0.0219
$FuzzyEn_{Hb}$	0.0145	0.0106	0.0556	0.0408
${}^{*}FuzzyEn_{Total}$	0.0708	0.0051	0.0243	0.0243
${}^{*}FuzzyEn_{concat}$	0.0219	0.0078	0.1735	0.0078
$DistEn_{HbO}$	0.6658	0.1735	0.1147	0.1619
$DistEn_{Hb}$	0.1272	0.0115	0.0709	0.2209
$DistEn_{Total}$	0.0871	0.0501	0.1411	0.0874
$DistEn_{concat}$	0.0408	0.1619	0.0118	0.0408

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
