# Peer review of "fNIRS Complexity Analysis for the Assessment of Motor Imagery and Mental Arithmetic Tasks"

_entropy, 2020, doi:10.3390/e22070761_

Round 1

Reviewer 1 Report

All comments are properly addressed.

Author Response

Dear reviewer,

We would like to extend our gratitude for your review. Having considered your critique, we have worked to improve the writing quality of the subsequently submitted manuscript which you shall find in the next resubmission.

Cordially,

Ameer Ghouse, Mimma Nardelli, and Gaetano Valenza

Reviewer 2 Report

General comments:

The authors reasonably addressed my questions given in the last review. However, I have another question, which was not appropriate since the layout of all optical probes was not given. Now, the authors provided Fig. 1, based on my request, showing all locations of the 36 fNIRS channels. These 36 channels covered only prefrontal cortex, limited occipital cortex, and both side of sensorimotor regions. Many cortical regions were uncovered by the fNIRS channels, and thus undetected.

The authors should mention in the manuscript for the following two points:
(1) all the quantified metrics (e.g., Mean oxy-, deoxy-, and total Hemoglobin SampEn and DistEn) in the uncovered/undetected cortical regions were derived based on interpolation.

(2) Because of dependence of interpolation for uncovered/undetected cortical regions, the authors should address potential errors or inaccurate/false topographic maps for Figs. 4-8.

Author Response

Dear reviewer,

We would like to extend our gratitude for your thorough review and your further suggestions to improve the manuscript. Each of your points were carefully considered and we hope the subsequent manuscript satisfies the standards of your critique. Please find below a point by point response to your criticism.

Cordially,

Ameer Ghouse, Mimma Nardelli and Gaetano Valenza

1.) all the quantified metrics (e.g., Mean oxy-, deoxy-, and total Hemoglobin SampEn and DistEn) in the uncovered/undetected cortical regions were derived based on interpolation.”

Response: Thanks for bringing this to our attention. We strongly agree that such methodology needs to be clarified in the manuscript for an honest interpretation of the results obtained. To that extent, we added a description of the interpolation performed in section 2.5 as follows:

Cortical regions in the topographic maps that are not covered by the optodes as seen in figure 1 are inferred using a bilinear interpolation.”

2.) “Because of dependence of interpolation for uncovered/undetected cortical regions, the authors should address potential errors or inaccurate/false topographic maps for Figs. 4-8.”

Response: We agree that this is critical to mention in the discussion in order to illuminate the potential flaws in the analysis that arise from the dataset used and to give direction for further research. To address that, we added the following commentary to the discussion section 4:

“In light of these significant results, it is important to mention that a bilinear spatial interpolation was performed on the topographic maps as mentioned in section 2.5, thus there could be errors in drawing the true cortical location of activity. In order to better pinpoint the true cortical location of specific activity, it would be important to use a higher density fNIRS cap in future experiments.”

This manuscript is a resubmission of an earlier submission. The following is a list of the peer review reports and author responses from that submission.

Round 1

Reviewer 1 Report

In this paper, the authors used entropy analysis to investigate the differences and similarities in hemoglobin concentration changes time series obtained using fNIRS. The authors used an already published dataset obtained from 29 subjects during motor imagery and mental arithmetic tasks. In comparison with standard methods such as the mean value of hemoglobin, the authors showed that entropy analysis uncovers potentially meaningful activation brain areas

This paper is very well structured. However, paper language needs to be further improved. The followings are my comments to improve the manuscript.

1) The entropy analysis is not new in the fNIRS analysis. The authors should show the significance of their studies in light of the following studies.

Keshmiri S, Sumioka H, Okubo M and Ishiguro H (2019) An Information-Theoretic Approach to Quantitative Analysis of the Correspondence Between Skin Blood Flow and Functional Near-Infrared Spectroscopy Measurement in Prefrontal Cortex Activity. Front. Neurosci. 13:79.

Keshmiri S, Sumioka H, Yamazaki R and Ishiguro H (2018) Differential Entropy Preserves Variational Information of Near-Infrared Spectroscopy Time Series Associated With Working Memory. Front. Neuroinform. 12:33

2) Figure 1 is not meaningful. The authors may remove it or add a block diagram of the whole process using in this manuscript.

3) What was the order of the Butterworth lowpass filter? The authors should justify the selection of cutoff frequency of 0.6Hz as this will not remove respiration and Mayer physiology.

4) What about motion artifacts? Furthermore, as the filters used in this study cannot eliminate task-related skin blood flow changes, therefore the authors should add comments on skin blood flow changes.

5) What is meant by the epoch in this study? Why the whole data was not converted into hemoglobin changes?

6) This statement “Each channel at each activity block is referenced to the mean of the previous 5s,” is not clear. The authors should more elaborate on the procedure used for the extraction of features.

7) How the authors selected the following “in this study it is set to 0.2* sigma_x”.

8) For the statistical analysis, the normality test was carried out or not?

Reviewer 2 Report

General comments:

This paper aims to characterize nonlinearities in fNIRS signals as assessment for motor imagery and mental arithmetics tasks. Specifically, the authors wished to investigate entropy of hemoglobin concentration time series obtained from fNIRS signals in order to determine differences and similarities between nonlinear and complexity analyses methods with standard mean hemoglobin value analysis of functional activation. The authors used publicly available data from 29 subjects taking motor imagery and mental arithmetics tasks for their purposed study. While the idea and goals were interesting and novel, the manuscript has several severe weaknesses to this reviewer. Overall, the manuscript is very hard to follow; also, it is lack of physiological interpretation and validation. Some details are given below. Thus, this reviewer’s recommendation for this manuscript is "major revision" or “Rejection.” 

  1. The manuscript did not provide any detailed information on how the fNIRS probes along with EEG electrodes were placed/arranged on subjects’ head. So, it is difficult to know where or which cortical regions the entropy metrics were covered.
  2. Along the similar line, it is unclear how the entropy metrics, such as SampEn, FuzzyEn, and DistEn, were calculated globally or for certain optodes. There was no place to mention/state the methodology. However, figures 3, 4, 5, and 6 show topographical maps, but there was no place giving information on how they quantify such channel-wise entropy metrics.
  3. How do the authors validate their nonlinearity results nonlinearities that are different from the conventional fNIRS signals? For example, the following paragraph was given in Section 3.2:

“… On the other hand, DistEn estimates on deoxyhemoglobin signal had significant changes between tasks over the somatosensory cortex that were not exhibited in the mean estimate. On the total hemoglobin signal, both SampEn and DistEn unraveled further information that mean estimates could not, where DistEn exhibite significant changes over the occipital area, and SampEn over the parietal area. In the concatenated topology, DistEn and SampEn exhibited different subsets of cortical activations.”

  • How are the authors prove their calculations correct while their results are different from those using the conventional method?
  • How did they perform concatenated topology? There was no place explaining or describing such operation.
  • Assuming the authors’ calculations are correct, what are the physiological meaning of these differences?
  • Again, how could the authors create such a topographical map without giving the actual optode locations?
  1. The comments or questions given in Point 3 can be held for Figs. 4-7. Some of the colormap topoplots seem to be very noisy. How would the authors confirm that such noisy data were derived from true physiological signals, not from low signal-to-noise ratios.